# The Multifaceted Roles of the Tumor Susceptibility Gene 101 (TSG101) in Normal Development and Disease

**DOI:** 10.3390/cancers12020450

**Published:** 2020-02-14

**Authors:** Rosa-Maria Ferraiuolo, Karoline C. Manthey, Marissa J. Stanton, Aleata A. Triplett, Kay-Uwe Wagner

**Affiliations:** 1Department of Oncology, Wayne State University School of Medicine and Tumor Biology Program, Barbara Ann Karmanos Cancer Institute, 4100 John R, EL01TM, Detroit, MI 48201, USA; ferraiur@karmanos.org; 2Eppley Institute for Research in Cancer and Allied Diseases, University of Nebraska Medical Center, 985950 Nebraska Medical Center, Omaha, NE 68198-5950, USA; kbaldus@coastal.edu (K.C.M.); mstanton3@unl.edu (M.J.S.); atriplett@unmc.edu (A.A.T.)

**Keywords:** TSG101 protein, cell death, cell survival, gene deletion, knockout, mouse, mutagenesis, oncogenes, transgenes

## Abstract

The multidomain protein encoded by the *Tumor Susceptibility Gene 101* (*TSG101*) is ubiquitously expressed and is suggested to function in diverse intracellular processes. In this review, we provide a succinct overview of the main structural features of the protein and their suggested roles in molecular and cellular functions. We then summarize, in more detail, key findings from studies using genetically engineered animal models that demonstrate essential functions of TSG101 in cell proliferation and survival, normal tissue homeostasis, and tumorigenesis. Despite studies on cell lines that provide insight into the molecular underpinnings by which TSG101 might function as a negative growth regulator, a biologically significant role of TSG101 as a tumor suppressor has yet to be confirmed using genuine in vivo cancer models. More recent observations from several cancer research teams suggest that TSG101 might function as an oncoprotein. A potential role of post-translational mechanisms that control the expression of the TSG101 protein in cancer is being discussed. In the final section of the review, we summarize critical issues that need to be addressed to gain a better understanding of biologically significant roles of TSG101 in cancer.

## 1. Identification of the *Tumor Susceptibility Gene 101* (*Tsg101*)

In the mid-1990s, the laboratory of Stanley Cohen at Stanford University conducted an insertional mutagenesis screen on immortalized mouse embryonic fibroblasts (NIH3T3 cells) to identify candidate tumor suppressor genes [1]. The functional knockout of both copies of a targeted locus was achieved through the expression of antisense transcripts from a retroviral gene search vector in an LAP348 transactivator-mediated manner. Using this approach, Li and Cohen identified more than 20 clones that exhibited anchorage-independent growth potential in soft agar, and one of these clones, called SL6, was expanded into a cell line. The transformed phenotype of these cells could be reversed when the expression of the antisense transcript was terminated through Cre recombinase-mediated excision of the transactivator transgene. Following identification of the insertion site of the gene search vector in SL6 cells and cloning of the extended cDNA from an expression library, the authors named the transcript originating from the targeted locus *Tumor Susceptibility Gene 101* (*Tsg101*). Designating this gene as a TSG seemed appropriate at the time since the exogenous expression of the *Tsg101* cDNA in both antisense and sense orientation resulted in a transformed phenotype. In 1997, *TSG101* had become more widely known as a tumor suppressor when it was reported that this gene is mutated, or its expression is lost in a significant subset of sporadic breast cancers [2]. These findings could not be confirmed [3,4,5,6], and it was also demonstrated later using immunoblot that ‘TSG101-deficient’ SL6 cells still express the TSG101 protein [7]. Regardless, TSG101 is still assigned as a tumor suppressor or negative growth regulator in online NCBI Resources and other internet databases.

## 2. *TSG101* is a Housekeeping Gene

The coding sequences of the mouse and human *TSG101* mRNAs are 86% identical on the nucleotide level [2]. Sanger sequencing data generated by our team revealed that the genomic organization of the *Tsg101* locus is highly conserved between mouse and human [8], and the mRNA transcripts span across 10 exons and not six as previously reported. The revision of the genomic structure of *TSG101* implied that the majority of truncated transcripts that were observed in cancer and nonmalignant tissues are true alternative splice products that originate from exon skipping, rather than aberrant transcripts from cryptic splice sites as proposed earlier [2,9,10,11]. The genomic sequencing results also demonstrated that the translation of the protein starts precisely with the known Kozak consensus motif [12]. The 5′ region preceding the first exon is typical for housekeeping gene promoters as it lacks TATA and CAAT boxes, and the highly GC-rich sequence contains several consensus sites for Sp1, AP2, and GAPBF2 [8]. In support of this notion, *TSG101* is expressed in all tissues and cell types [2,8], and the analysis of expressed sequence tags (ESTs) revealed that the *Tsg101* mRNA is already present in 1-cell and 2-cell stage mouse embryos. The expression of *Tsg101* in germ and stem cells may also explain the origin of a processed *Tsg101* pseudogene in the mouse genome, which made it challenging to identify the actual *Tsg101* locus and isolate genomic DNA sequences for the construction of gene targeting vectors to generate knockout mice [8,13,14,15]. Despite ubiquitous expression in all tissues, it might be worth noting that the highest *Tsg101* mRNA levels were observed in the brain and the lactating mammary gland [8].

The designation of *Tsg101* as a housekeeping gene had several implications. First, a complete knockout of *Tsg101* might cause early embryonic lethality [8]. More importantly, significant variations in high or low protein expression levels in normal tissues or cancer cells are likely a consequence of post-transcriptional or post-translational mechanisms. In the postgenomic era, this is an important fact to consider since *mRNA* expression levels from microarray and RNA-sequencing data are frequently being used to judge the importance of genes in cancer development and patient survival. As discussed later in this review, a tight post-translational control of the TSG101 protein level that balances variations in mRNA expression also imposes challenges for the generation of genetically engineered models to assess the effects of TSG101 gain- or loss-of-function in normal organogenesis and cancer development.

## 3. *TSG101* Encodes a Multidomain Protein

The human and mouse *TSG101* transcripts encode proteins of approximately 50 kDa in size with more than 94% similarity. The TSG101 protein contains several conserved domains (Figure 1). The N-terminal region is a ubiquitin-conjugating enzyme E2 variant (UEV) domain, which has been shown to bind ubiquitin but it lacks enzymatic activity due to the absence of the active site cysteine residue that is required for the transfer of ubiquitin [16,17,18,19]. Based on this unique structural feature, TSG101 was suggested to function as a negative regulator of ubiquitin-mediated protein degradation [16] as well as a mediator for the intracellular movement of ubiquitinated proteins [19]. The UEV domain also contains a hydrophobic groove that facilitates the association of TSG101 with polypeptides that contain specific PTAP or PSAP amino acid motifs such as regulatory proteins for intracellular trafficking and retroviral proteins [20,21,22,23]. Other conserved structural features of TSG101 are a proline-rich region (PRD) that is typically found in surface proteins and transcription factors [2] as well as a coiled-coil (CC) domain that has been shown to interact with stathmin [24]. The C- terminal end of TSG101 was designated as a ‘steadiness box (SB)’ because of its critical role for the post-translational autoregulation of steady-state levels of the TSG101 protein [25]. Interestingly, TSG101 contains an intrinsic PTAP amino acid motif located between the CC and SB that might modulate the binding of proteins to the N-terminal UEV domain of TSG101 [26].

## 4. The TSG101 Protein Mediates Diverse Intracellular Processes

On the subcellular level, most of TSG101 was observed to reside in the cytoplasm regardless of the cell cycle stage [27]. During the late S phase, a fraction of the protein can be detected in the nucleus, and TSG101 co-localizes with the spindle apparatus during mitotic cell division [5,27,28]. The temporal distribution of TSG101 within several cellular compartments is indicative that this multi-domain protein facilitates a variety of molecular and biological processes, which include the regulation of transcription [29,30], cell proliferation, and division [27,31,32], as well as the aforementioned ubiquitination and intracellular movement of other proteins.

Two reports from the laboratories of Duane Jenness and Scott Emr [33,34] were first to classify *TSG101* as the mammalian ortholog of the class E *Vacuolar protein sorting 23* (*VPS23*) in *Saccharomyces cerevisiae,* which is also known as *STP22*. Subsequent studies revealed that TSG101/VPS23 is a central member of a subset of VPS class E proteins (VPS23, VPS28, and VPS37) that form the Endosomal Sorting Complex Required for Transport I (ESCRT-I). As part of this complex, TSG101 is suggested to selectively bind ubiquitinated cargo proteins and direct their sorting into multivesicular endosomes [19]. In line with these findings, it was reported later that the downregulation of TSG101 by RNA interference causes a formation of multi-cisternal early endosomes and defects in protein sorting. These intracellular abnormalities may impair the downregulation of the endocytosed, ligand- bound Epidermal Growth Factor Receptor (EGFR) [26,35,36,37], which consequently may enhance receptor signaling in early endosomes. This suggested role may provide a molecular mechanism by which TSG101 could act as a tumor suppressor, and it was reported that ‘TSG101-deficient’ SL6 cells exhibited alterations in EGF trafficking that may have caused elevated levels of active MAP kinases following EGF stimulation [34]. Nonetheless, these early reports should be viewed with some caution since it was revealed later that the ‘TSG101-deficient’ SL6 cell line did not show any substantial reduction in the steady-state level of the TSG101 protein as determined by immunoblot [7]. More importantly, it was also demonstrated that cells completely deficient in TSG101 (i.e., genomic knockout) exhibited a significant reduction in EGFR and ERBB2 [38], and, as discussed later, there is no experimental evidence to date that would suggest that deficiency in TSG101 is sufficient to cause neoplastic transformation and cancer [7,14,15,39].

Insight into the multifaceted functions of TSG101 as a member of ESCRT-I was gained from a series of studies on the association of TSG101 with retroviral proteins and their intracellular movement [40]. TSG101 is being recruited and delivered to viral assembly sites by interacting with group-specific antigen (Gag), which has a PTAP amino acid motif in its C-terminal p6 region that can associate with the PTAP binding grove within the N-terminal UEV-domain of TSG101 [18,21,22,41,42]. The ubiquitination (Ub) of Gag permits another mode of interaction with a different region in the UEV-domain of TSG101 [23,43].

The specific functions of TSG101 in retroviral particle assembly and budding might be exemplary for a broader role in the biosynthesis and release of exosomes or, as now generally defined, secreted extracellular vesicles (EVs). Using a proteomic approach, Thery et al. [44] were first to identify the TSG101 protein within exosomes, and TSG101 is now being employed as one of the standard markers for the purification of EVs [45]. In a newer study, Colombo and coworkers [46] reported that selected ESCRT proteins modulate EV biogenesis, and the downregulation of TSG101 resulted in reduced secretion of exosomes. Since TSG101 seems to play a general role in EV assembly and release in mammalian cells and retroviruses appear to hitchhike on these EV mechanisms, it should be evident that membrane-binding determinants in the matrix domain of retroviral Gag proteins may not be the main mode of (mis)directing TSG101 and its cargoes to the cell membrane. Therefore, the specific control mechanisms by which TSG101 may traffic its ubiquitinated cargos to the cell membrane, as opposed to degradative compartments in the interior of a cell, remain to be elucidated.

In summary, the TSG101 protein is suggested to mediate a variety of intracellular processes, but the majority of published articles only highlight its proposed roles in endosomal trafficking. Notably, many of the reports that solely focus on the molecular, biochemical, and intracellular functions of TSG101 emphasize the importance of their specific findings for diseases like cancer and neurodegeneration, but the proposed mechanisms are rarely validated in genetically defined in vivo disease models or in primary human tissue samples. Consequently, the biological relevance of these multifaceted functions of TSG101 for normal development, differentiation, and tissue homeostasis, as well as potential roles of TSG101 in cancer are poorly defined. On the experimental level, most published studies that examined the intracellular roles of TSG101 relied exclusively on RNA interference, overexpression of TSG101 mutants, or the use of chemical compounds that disrupt entire cellular processes. CRISPR/Cas-based targeted knockouts or other gene editing methods to introduce mutations into the endogenous *TSG101* locus have not been widely employed. As reviewed in the next section, the phenotypes of defined TSG101 genomic knockout models deviate from observations in cell lines where TSG101 is only knocked down with siRNAs. It is unclear to date which of the known molecular mechanisms, if any, are responsible for the indispensable functions of TSG101 for the proliferation and survival of mammalian cells.

## 5. A Knockout of TSG101 Causes Cell Cycle Arrest and Cell Death

From the analysis of the *Tsg101* gene promoter sequence and expression profiling, we predicted that a conventional knockout of this gene causes embryonic lethality [8]. Indeed, Ruland and coworkers [14] were first to demonstrate that mice homozygous deficient in exons 8 and 9 fail to develop past day 6.5 of embryogenesis. TSG101 knockouts do not form a mesoderm, and they are smaller as a result of a defect in cell proliferation, which was caused by the nuclear accumulation of p53. In line with this notion, embryos that are doubly deficiency in TSG101 and p53 were able to gastrulate, but they survived for only two additional days. Our team developed a Cre/lox-based conditional knockout mouse model that allows for a temporally and spatially controlled deletion of *Tsg101* in germ cells and differentiated tissues during embryonic and postnatal development as well as in cultured cells that are primary, immortalized, or tumorigenic [7,15,39]. The conditional deletion of the promoter and first exon with Cre recombinase resulted in the complete absence of the TSG101protein [7]. As controls to the genomic knockout, we examined the level of TSG101 in the ‘TSG101-deficient’ SL6 cell line described earlier and were surprised that these cells exhibited only a marginal reduction in the TSG101 protein compared to their parental NIH3T3 cells. Similar to the report by Ruland et al., we observed that a complete knockout of TSG101 causes a p53-dependent cell cycle arrest, but the deletion of *p53* did not extend the survival of TSG101-deficient cells [7,39]. Re-expression of exogenous, epitope-tagged TSG101 was sufficient to restore normal proliferation and survival of cells that lacked both endogenous copies of *Tsg101*. More interestingly, we noted that individual rescue clones expressed exogenous TSG101 at the same levels as the endogenous protein in parental controls [7], supporting earlier reports that the amount of the TSG101 protein in a particular cell type is tightly controlled [25]. We have used this knockout-rescue approach to validate that expression of the human *TSG101* cDNA can fully restore normal growth of mouse cells that lack endogenous TSG101, suggesting that both mammalian proteins function in an identical manner. In contrast to the full-length cDNA, expression of *TSG101* mRNA splice variants or deletion mutants lacking individual domains of TSG101 did not rescue the deleterious knockout phenotype (Stanton and Wagner, unpublished) [47].

In contrast to mutants that lack exons 8 and 9, the Cre-mediated excision of the promoter and exon 1 of *Tsg101* in the germline resulted in an earlier embryonic lethality prior to or during implantation [15]. A tissue-specific knockout of TSG101 in mammary epithelial cells of lactating female mice using Cre recombinase under control of the *Whey Acidic Protein* gene promoter (WAP- Cre) demonstrated that TSG101 is equally required for cell survival and tissue homeostasis in adult animals. Secretory epithelial cells that have completed their functional differentiation show a significantly reduced proliferation rate and it is therefore evident that crucial functions of TSG101 for cell survival are independent of its suggested roles in cell division. Re-expression of exogenous TSG101 in the lactating mammary gland restores normal mammary gland development and lactation [48]. Similar to the results from cell line studies, the survival of TSG101-deficient epithelial cells in the mammary gland could not be rescued by a knockout of p53 [39].

TSG101 conditional knockout mice were also utilized to examine biologically relevant functions of TSG101 for normal tissue homeostasis in organ systems other than the mammary gland. Using a cardiac-specific and tamoxifen-inducible knockout of TSG101, Essandoh et al. [49] recently demonstrated that complete deficiency in TSG101 was lethal within one week after deleting both copies of the gene in adult mice. In contrast, animals with a 50% knockdown of TSG101 through the expression of an shRNA construct were able to survive, but they exhibited defects in exercise-induced cardiac hypertrophy. This suggests that optimal expression of TSG101 is required for heart muscle growth under specific physiological conditions, and interestingly, reduced levels of TSG101 can still be lethal when the mice are being challenged with endotoxin-triggered myocardial injury [50]. On the mechanistic level, exercise-induced cardiac growth is being mediated by IGF-1R signaling through AKT, and a knockdown of TSG101 caused a defect in the recycling of the IGF1 receptor, possibly due to the downregulation of RAB11a and FIP3. These findings may support an observation from our team that shows that the knockout of TSG101 causes the dissociation of early endosomes from the recycling endosome (Figure 2). Then again, the mechanism does not explain the cell death caused by a complete deficiency in TSG101 and specifically the lethal phenotype of mice with the heart-specific knockout. In stark contrast to the TSG101 conditional knockout mice, the Cre-mediated deletion of the IGF-1R in cardiomyocytes had no noticeable effect on normal heart development [51].

Like the mammary epithelium and cardiomyocytes, the deletion of TSG101 in oligodendroglia was reported to cause apoptosis, demyelination, vacuolation, and severe spongiform encephalopathy. Mice lacking TSG101 in parts of the central nervous system developed a tremor and had significantly reduced body weight by 8 weeks of age [52]. In a parallel study, the authors showed that conditional knockout mice deficient in the endosomal protein RAB7 did not show any histopathological abnormalities in the CNS, suggesting that the severe phenotypes associated with TSG101 deficiency are not due to any defects in late endosomal, lysosomal, or autophagolysosomal functions as proposed earlier from cell line studies. The authors concluded that the vacuolation in the brains of TSG101-deficient animals is likely a secondary consequence.

In summary, the complete loss of TSG101 triggers an accumulation of p53 and a cell cycle arrest at the G1 phase followed by cell death, which is independent of p53 function and proliferation. This finding is not unique to mammals and can be extended to other vertebrate species. A knockdown of TSG101 with antisense morpholinos causes a growth delay and lethality of zebrafish embryos [53]. On a mechanistic level, it was validated that p21^Cip1^ is a downstream mediator of the p53-dependent cell cycle arrest in response to the deletion of *Tsg101* [39,53]. Although it was previously proposed that TSG101 controls the levels of MDM2 and vice versa [54], we demonstrated that deficiency in *Cdkn2a* (p16^Ink4a^/p19^Arf^ knockout) did not alter the ability of MDM2 to sequester p53 and restore the p21-mediated G1 arrest. The collective results obtained from the analysis of *Tsg101/p53*, *Tsg101/p21*, and *Tsg101/Cdkn2a* conditional double-knockout cell lines that were generated in our team suggested that a biologically significant interaction of TSG101 with the MDM2/p53 circuit seems unlikely. The p53/p21-controlled G1 arrest might, therefore, be an indirect consequence of an activation of stress response pathways that are being triggered by the knockout of TSG101 [39].

## 6. TSG101 Knockout Cells Are Stressed and Undergo Autophagy Prior to Cell Death

Fibroblasts with a conditional knockout of TSG101 exhibit several characteristics of stressed cells: a) phosphorylation of mitogen-activated protein (MAP) kinases independent of growth factor stimulation, b) widespread redistribution of actin filaments, and c) induction of autophagy prior to cell death [38]. TSG101 knockout cells possess greatly enlarged lysosomes that were enriched with the autophagy-related protein LC3. Unlike previous observations in SL6 cells [34], TSG101 is not required for the routing of biosynthetic cargo from the Golgi and delivery of hydrolases to lysosomes. Cathepsin D is transported to and functional within the distended vesicles of TSG101 knockout cells [38]. While re-expression of exogenous TSG101 can revert these intracellular processes, treatment of TSG101 knockout cells with the PI3 kinase inhibitor 3-methyladenine (3MA) led to accelerated cell death, supporting the notion that TSG101 knockout cells utilize autophagy as a survival mechanism prior to their ultimate death [38].

Although the precise cellular process that causes TSG101 knockout cells to die is still unknown, one important consequence that may trigger an intense stress-response is growth factor deprivation. Unlike previous observations that a siRNA-mediated knockdown of TSG101 resulted in enhanced recycling or retention of the EGFR within early endosomes [55], the defined conditional knockout of the *Tsg101* gene causes a substantial decline in the steady-state levels of the EGFR and ERBB2 [38]. This may have been the result of the dissociation of early endosomes from recycling endosomes (Figure 2). This finding is in line with an earlier report showing that TSG101 can associate with RAB11-family interacting proteins FIP3 and FIP4 [56], and this interaction might be crucial for the recycling of receptor tyrosine kinases (RTKs), including IGF-1R as discussed earlier [49]. It remains to be determined whether the reduced expression of RTKs and cellular stress responses are initiating factors or secondary events that cause TSG101 knockout cells to die.

An important question is whether the proposed significant roles of TSG101 in the endocytic trafficking of ubiquitinated cargo proteins, which include RTKs, play any role in the cell survival functions of TSG101. As reviewed earlier, key functions of TSG101 in endocytic trafficking are mediated by the UEV domain of TSG101, which is able to independently bind ubiquitin and P(S/T)AP motif-containing proteins at different sites. Pornillos and coworkers [20] demonstrated that distinct point mutants of TSG101 significantly impair PTAP (M95A) or ubiquitin (N45A) binding. Given the suggested significance of these amino acids for the endosomal trafficking of viral proteins, we were surprised to find that, similar to our published knockout-rescue experiment with wildtype TSG101 [7,38], the re-expression of hemagglutinin (HA) epitope-tagged TSG101 M95A and N45A mutants was able to restore the proliferation and survival of TSG101 conditional knockout cells (Figure 3A, left panel). The lack of both endogenous *Tsg101* alleles in surviving knockout rescue clones was confirmed by Southern blot (Figure 3A, right panel). Importantly, the levels of both mutant proteins were similar to endogenous TSG101 in the parental controls (*Tsg101^fl/fl^*) as well as knockout rescue clones expressing HA-tagged exogenous, wildtype (wt) TSG101 (Figure 3B). Given that TSG101 levels in normal cells are being maintained within a narrow physiological range though post-translational mechanisms, we conclude that both mutants are functionally equivalent to the wildtype protein in those cellular processes where TSG101 is crucial for cell survival. The results from the knockout rescue experiments with TSG101 mutants suggest that critical functions of TSG101 in growth and survival do not seem to be strictly dependent on an effective binding of ubiquitin or PTAP-containing proteins to its N-terminal UEV domain. Therefore, it might be possible to target regions within the UEV domain of TSG101 that are crucial for retroviral assembly and release without causing deleterious side effects that are related to other, biologically more significant functions of TSG101.

## 7. Lack of Clear Evidence for a Tumor-Suppressive Role of TSG101 in Mammalian in vivo Models

Despite a wealth of information about multifaceted intracellular functions of TSG101, it is still a conundrum why a knockout of the mammalian *Tsg101* gene causes cell cycle arrest and cell death instead of accelerated growth and neoplastic transformation, which would support its proposed function as a tumor suppressor. This is also the case for other vertebrate animal models such as zebrafish [53]. The essential role of TSG101 for cell survival seems to apply to all mammalian cell types that have been examined thus far including those that lack the bona fide tumor suppressor genes *p53* and *Cdkn2a.* TSG101 is also essential for the growth of neoplastic cells that are capable of forming tumors in mice [7,15,39]. Neither haploinsufficiency nor a complete deletion of both *Tsg101* alleles in somatic tissues of genetically engineered mice, such as the mammary epithelium, led to the development of benign neoplasms or cancer [14,57]. Moreover, we did not observe any mammary tumors in females where *Tsg101* was deleted in the mammary epithelium that lacked both copies of the tumor suppressor p53 [39]. Instead of promoting mammary tumorigenesis, complete deficiency in TSG101 prevented the onset of mammary neoplasia in mice that overexpressed the ERBB2 oncogene [58]. This was likely a consequence of the growth inhibition of an epithelial subtype in the mammary gland that is susceptible to ERBB2-induced transformation. Deletion of both alleles of *Tsg101* in established, ERBB2-transformed mammary cancer cells causes cell death (Triplett and Wagner, unpublished) [59]. These findings are similar to studies that showed that an effective downregulation of the TSG101 protein by RNA interference caused reduced proliferation and colony formation, as well as an impaired migration of human prostate (PC3), breast (MDA-MB-231, MCF- 7), and renal cancer cells (A-498 and 786-O) [60,61,62]. In SKOV-3 ovarian cancer cells, the siRNA-mediated knockdown of TSG101 led to a cell cycle arrest at the G2/M phase prior to apoptosis [63]. Silencing the expression of TSG101 in Huh7 human hepatocellular carcinoma cells resulted in abnormal actin filaments, growth arrest, and induction of autophagic cell death [64], which, as reviewed earlier, are similar phenotypic abnormalities that we observed in TSG101 conditional knockout fibroblasts [38]. On the mechanistic level, the reduced migratory features of cancer cells with a knockdown of TSG101 might be a consequence of impaired trafficking of c-SRC to focal adhesions, as well as a loss of Focal Adhesion Kinase, MAPK, and STAT3 activation [61,65].

In support of a tumor-suppressive role of TSG101, Moberg and coworkers [66] reported that lack of the *TSG101* ortholog *Erupted* in *Drosophila* causes excessive cell growth, albeit in a *non- cell- autonomous* manner through the expression of the Notch target gene *Unpaired*, which is the secreted ligand that stimulates JAK/STAT signaling in the fly. It is obvious that a mechanism of a non-cell-autonomous role of TSG101 in growth suppression in the fly does not correspond to the phenotypes of a knockdown of TSG101 in normal and neoplastic cells of human and mouse origin. Although the proposed function of *Erupted* as a paracrine growth suppressor seems interesting, it needs to be explained why the phenotype of the *Erupted*-deficient fly is very different from a knockout of *dVps28*, which results in early embryonic lethality [67]. TSG101 and VPS28 bind directly to each other in mammalian cells [68], and more importantly, the depletion of TSG101 in human cell lines lead to a significant downregulation of the VPS28 protein [35,69]. The divergent phenotypes of *Erupted* and *dVps28* in mutant flies might suggest that the mechanism that controls the post- translational cross-regulation of the levels of TSG101 and VPS28 in mammalian cells may not exist in *Drosophila*. Regardless of the nature of these discrepancies, we and others have never observed any palpable tumors in TSG101 conditional knockout mice where one or both copies of *Tsg101* were deleted in a mosaic fashion in selected cell types with Cre recombinase. The same was true for a morpholino-mediated knockdown of TSG101 in zebrafish [53]. Hence, there is no conclusive experimental evidence to date using vertebrate models to suggest that TSG101 functions as a paracrine growth suppressor.

In conclusion, the various genetically engineered animal models, as well as observations from human cancer cell lines where TSG101 was knocked down with siRNAs, do not support a tumor- suppressive role of its encoded protein, either in a cell-autonomous manner or by means of juxtracrine or paracrine signaling. As stated in the opening paragraph of this review, initial reports about a loss of the *TSG101* gene in human cancers could not be confirmed. Moreover, alternative splice variants of *TSG101*, which have been subject to intense investigations in the late 1990s, can only be detected by nested RT-PCR, and they are also present in normal tissues [9]. Most of the splice variants do not encode any functional protein and are not expressed at levels that are comparable to that of the full-length mRNA. Using Northern blot, only one strong band corresponding to the full- length message has been detected in normal tissues and mammary cancers [2,6,8]. As mentioned earlier, the lack of a significant reduction of the TSG101 protein in the original ‘TSG101-deficient’ SL6 cells should have raised questions about its assigned tumor-suppressive role as well as the validity of subsequent findings when these cells were used for mechanistic studies on the molecular functions of TSG101 without proper validation of the TSG101 protein expression using quantitative methods. To our knowledge, the transforming ability of a reduced expression of TSG101 by means of an antisense construct has never been repeated in normal human cells. Unless it can be demonstrated using genetically defined models that lowering the steady-state expression of TSG101 can trigger the onset or progression of cancer, the molecular mechanisms that were published in support of the elusive tumor-suppressive function of TSG101 should be viewed with caution.

## 8. TSG101 Might Function as an Oncoprotein

Investigations from various research teams including our own have shown that TSG101 is overexpressed rather than lost in a significant subset of malignancies, including breast, lung, thyroid, ovarian, and colon cancers [48,63,70,71,72,73]. According to publicly available information in the Human Protein Atlas (www.proteinatlas.org), most human cancers exhibit a moderate cytoplasmic and membranous immunoreactivity against TSG101, and the intensity of the staining might serve as a prognostic marker for selected cancer types such as hepatocellular and renal tumors. Using immunohistochemistry staining on a limited set of primary human breast cancers and normal breast tissues, our team observed that TSG101 is elevated in 50% of invasive breast cancers [48]. However, a more detailed analysis of the levels of TSG101, in particular breast cancer subtypes, using quantitative methodologies is still warranted. To illustrate this fact, we conducted a gene expression analysis on more than 3900 breast cancer cases using KM plotter (kmplot.com). Similar to the information from the Human Protein Atlas, a high or low mRNA expression of *TSG101* across all cases may not serve as a prognostic marker (Logrank *p* = 0.49, N = 3951). In contrast, higher levels of *TSG101* transcripts correlate significantly with reduced survival in luminal type A (Logrank *p* = 0.017, N = 2504) and in luminal type B (Logrank *p* = 0.0084, N = 1425) breast cancers (Ferraiuolo et al., manuscript in preparation). This implies that any studies that are designed to examine the biological significance of TSG101 and associated molecular mechanisms for a given tumor type (e.g., breast cancer) should be conducted on cell lines and animal models that resemble the main characteristics of a particular cancer-intrinsic subtype (e.g., luminal-type breast cancer models).

Similar to breast cancer, it has been reported that TSG101 is upregulated in human ovarian epithelial cells that express oncogenic HRAS or KRAS as well as in 70% of human ovarian carcinomas analyzed on tissue arrays [63,72,74]. While the normal human ovarian epithelium does not exhibit any significant expression, both low- and high-grade tumors were found to have elevated levels of TSG101. More importantly, the 5-year survival rate of patients is significantly lower (33%) when their cancer tissues exhibited higher levels of TSG101 compared to patients with a low expression of this protein (53%). These findings suggest that TSG101 might be a prognostic marker for poor overall survival in ovarian cancer [72].

The elevated expression of TSG101 in cancer compared to normal tissues may support the notion that TSG101 plays a role at particular stages of cancer initiation or progression. Experimental evidence that TSG101 overexpression might contribute to neoplastic transformation was provided in the inaugural paper by Li and Cohen [1]. This finding was validated in an independent study by Liu and coworkers [71], who isolated *TSG101* as a transforming oncogene when an expression library of a highly metastatic lung adenocarcinoma cell line was introduced into mouse NIH3T3 cells. To our knowledge, the transforming ability of TSG101 overexpression has not been assessed in any other cell lines, in particular, those of human origin. As mentioned earlier, TSG101 protein levels are being maintained within a relatively narrow range of a given cell type, which may complicate the generation of suitable overexpression models. A minor or moderate expression of exogenous TSG101 results in a compensatory downregulation of the endogenous protein on the post-translational level [25]. Possible mechanisms for this phenomenon will be discussed in the next section. In contrast, a sustained overexpression of TSG101 in primary and untransformed immortalized cells, and even in cancer cell lines, often results in cell death [27]. It is a conundrum why only a few cell types are able to tolerate significantly elevated levels of exogenous TSG101. For example, sustained overexpression of epitope-tagged TSG101 under control of the *Whey Acidic Protein* (*Wap*) promoter can be achieved in functionally differentiated epithelial cells of the lactating mammary gland [48]. TSG101 overexpressing epithelial cells did not undergo apoptosis despite the activation of the STAT3, which is normally associated with the induction of apoptosis. This may have been a consequence of active STAT5 downstream of prolactin signaling, which has been demonstrated to function as a potent survival factor that can effectively overwrite STAT3-mediated pro-apoptotic signals [75]. In another transgenic model, Essandoh and colleagues [49] achieved a high expression of TSG101 in cardiomyocytes under the regulation of the α-myosin heavy chain promoter (α-MHCp). Interestingly, excess levels of TSG101 in this organ lead to enhanced cardiac function and hypertrophy of the heart due to an increase in the size of individual cardiomyocytes rather than an increase in their number or in fibrosis. Malignant transformation was not reported, and this is not surprising since cancer of the heart is extremely rare and is known to originate from connective tissue.

In contrast to the heart-specific overexpression model, a subset of transgenic females (approximately 20%) that express exogenous TSG101 under control of the *Wap* gene promoter developed mammary tumors after a relatively long latency [48]. At the experimental endpoint of nearly two years, more than half of all surviving WAP-TSG101 transgenic females exhibited preneoplastic or inflammatory lesions in the mammary gland. Atypical hyperplasia and squamous metaplasia were far more common in the transgenics compared to age-matched wildtype controls. The long latencies in tumor formation in this model led us to conclude that TSG101 possesses weak oncogenic properties associated with cancer initiation. Nonetheless, it is currently unknown whether all epithelial subtypes in the mammary gland are equally susceptible to TSG101-associated neoplastic transformation. While the *Wap* gene promotor is suitable to drive a tissue-specific expression of genes almost exclusively to the mammary epithelium, a high activation occurs predominantly in functionally differentiated alveolar cells during pregnancy and lactation, and the majority of these cells die during post-lactational mammary gland remodeling following the weaning of the offspring. As such, exogenous TSG101 is not persistently overexpressed in most epithelial cells throughout the lifetime of a female mouse. It is therefore not known whether a persistently high expression of TSG101 in diverse epithelial subtypes of the mammary gland may cause an earlier onset of mammary cancer. To our knowledge, the WAP-TSG101 transgenic mouse strain is currently the only published in vivo model that demonstrated that TSG101 can function as a transforming oncogene that causes a sporadic occurrence of preneoplastic lesions and mammary cancer. Given the long latencies of tumor formation, it is evident that secondary genetic or epigenetic changes must occur that drive the progression of bona fide cancers in this model.

## 9. Post-Translational Control of TSG101 Expression and Its Potential Role in Cancer

As reviewed earlier, *TSG101* possesses characteristics of a housekeeping gene, and significant variations in the expression of its encoded protein between diverse cell types are, in part, a consequence of post-translational mechanisms. Given the stringent control of the TSG101 protein levels in a particular cell type, it is likely that the deregulated expression of TSG101 before or during neoplastic progression is primarily caused by an interference with its post-translational control mechanisms. The precise molecular processes that regulate the amount of TSG101 within cells are less well defined and have not been associated with any biologically relevant functions of TSG101 as an oncoprotein.

MDM2, LRSAM1/TAL, and MGRN1 are three E3 ubiquitin ligases that have been reported to control TSG101 levels in mammalian cells. Li and coworkers [54] have shown that when overexpressed in Saos-2 cells, TSG101 can interact with MDM2 and prolong the half-life of the ubiquitin ligase. In contrast, the authors also reported that MDM2 overexpression led to an accelerated decay of the TSG101 protein. Thus far, it has not been demonstrated that TSG101 levels are dependent on the functionality of MDM2 in cancers that are driven by this bona fide oncogene. If proven correct using in vivo cancer models or human specimens, the proposed mechanism would imply that TSG101 levels should be generally lower in cancers that overexpress MDM2 or lack p19^Arf^. Evidently, TSG101 may not function as an oncoprotein under these physiological conditions. In contrast, TSG101 may only exert oncogenic activities in the MDM2/p53 circuit when it is upregulated and able to stabilize MDM2 without being ubiquitinated and degraded by this ubiquitin ligase.

Using a yeast two-hybrid system, Amit et al. [76] identified TSG101-Associated Ligase (TAL) as a novel E3 ubiquitin ligase that ubiquitylates TSG101. TAL is also known as Leucine-rich repeat and sterile alpha motif-containing protein 1 (LRSAM1), and mutations in this gene are linked to Charcot– Marie–Tooth disease, which is a heterogeneous group of inherited motor and sensory neuropathies [77,78,79]. The N-terminal region of TAL, which contains a tandem PT/SAP motif, has been demonstrated to interact with TSG101 by binding to its UEV domain. Additionally, the central region of TAL associates with the C-terminal SB domain of TSG101 [76]. TAL is suggested to monoubiquitinate TSG101 at multiple lysine residues in the SB domain, which re-localizes the protein from being membrane-bound to the cytoplasm, thereby inactivating its sorting activities for receptor tyrosine kinases and viruses. In a subsequent study, McDonald and Martin-Serrano [80] reported that the TAL-mediated polyubiquitination of lysine residues in the C-terminus of uncomplexed TSG101 causes its proteasomal degradation. On a mechanistic level, the authors reported that the M95A point mutation in TSG101 abolished its binding to TAL. If this had been correct, we should have seen greatly elevated levels of epitope-tagged TSG101 in our genetically defined TSG101 knockout rescue clones that only express the M95A mutant protein compared to wildtype rescue clones (Figure 3). This, however, was not the case in any of the individual M95A rescue clones, supporting the previously stated notion that the interaction between TAL and TSG101 is more complex [76]. It is also likely that there are other intracellular mechanisms at play that control the level of TSG101 in the absence of TAL. Whether loss of TAL expression and functionality contributes to an upregulation of TSG101 in cancer is entirely unclear, and it has not been reported that LRSAM1 mutant mice, which exhibit only a very mild neuropathy phenotype with age [81], develop any tumors. According to the Human Protein Atlas, higher levels of TAL/LRSAM1 may serve as favorable prognostic markers in pancreatic and renal cancers, but it remains to be determined whether variations in the expression of TAL/LRSAM1 in selected cancer subtypes also correlate with changes in TSG101 protein levels.

Like the PTAP–PSAP double motif in TAL, the Mahogunin Ring Finger-1 (MGRN1) E3 ubiquitin ligase possesses a single, evolutionarily conserved PSAP amino acid sequence that can bind to the UEV domain of TSG101 [82]. MGRN1 monoubiquitinates TSG101 on multiple sites, and its siRNA- mediated depletion has been reported to impair the trafficking of EGFR to the lysosome and prolong signaling. In contrast, MGRN1 may not be required for viral budding [82,83]. MGRN1 knockout mice progressively develop spongiform neurodegeneration over the course of a year. While MGRN1 may not have a significant role in controlling the level of the TSG101 protein, this ubiquitin ligase might modify the trafficking and functionality of TSG101. By three months of age, brain tissues of MGRN1 knockout mice exhibited a build-up of insoluble, multi-ubiquitinated TSG101 that may not be correctly degraded in the proteasome and may interfere with its normal functions. If this model is correct, it could be possible that MGRN1 might mediate the oncogenic functions of TSG101. In support of this notion, the *MGRN1* locus has been reported to be amplified in osteosarcomas [84] and, according to information provided in the Human Protein Atlas, most cancer types, except gliomas, showed a moderate to strong cytoplasmic staining of the MGRN1 protein. Similar to TAL, it is currently unknown whether a deregulated expression of MGRN1 significantly contributes to the expression and oncogenic properties of TSG101 in cancer.

## 10. Conclusions

Although TSG101 is suggested to function in a variety of cellular processes, the vast majority of published reports over the past 20 years have highlighted specific molecular mechanisms by which TSG101 may control the intracellular trafficking of cargo proteins such as RTKs and viral particles. Key observations from these studies are often highlighted as proof of the molecular underpinnings by which TSG101 may function as a growth suppressor. Then again, the validity of the proposed mechanisms is rarely, if at all, tested in genetically defined disease models or in primary human cancer tissues. At present, there is no conclusive evidence from experimental models or human tissues in support of the notion that deficiency in TSG101 or a functional loss of the protein could contribute to the genesis and progression of cancer. It has been established that TSG101 may function as a protooncogene in certain cell types. In the context of a whole organism, the biological significance of the proposed multifaceted roles of TSG101 in developmental processes and normal tissue homeostasis is still unclear. None of the proposed molecular roles sufficiently explains why cells that completely lack TSG101 enter a cell cycle arrest and die. Many of the critical functions in the intracellular trafficking of cargos as well as the binding of factors that control the post-translational modification and functionality of the TSG101 protein have been attributed to two structural features, i.e., PTAP amino acid motif-binding and ubiquitin-binding domains. Future studies using mouse models with CRISPR/Cas9-mediated targeted knock-in mutations might show whether these two domains or additional amino acid residues of TSG101 have a biologically significant role in normal development and disease.

## Figures and Tables

**Figure 1 cancers-12-00450-f001:**
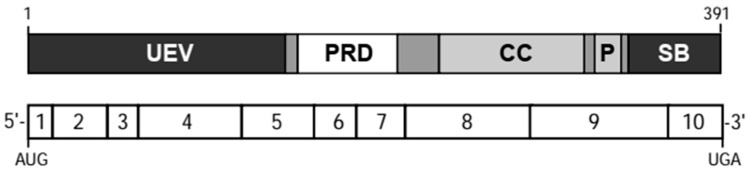
Schematic of the functional domains of the mouse TSG101 protein and their location within the 10 coding exons of the spliced mRNA. UEV, ubiquitin-conjugating enzyme E2 variant; PRD, proline-rich domain; CC, coil-coiled domain; P, conserved PTAP tetrapeptide motif; SB, steadiness box.

**Figure 2 cancers-12-00450-f002:**
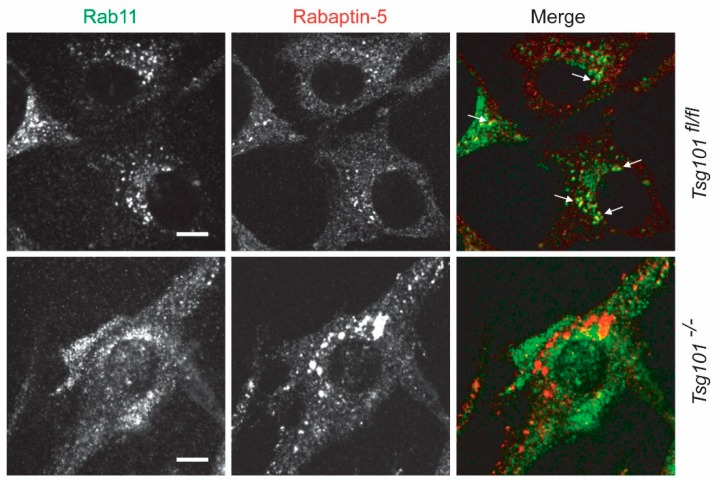
The knockout of TSG101 causes a dissociation of a subset of early endosomes from recycling endosomes Confocal microscopic images of TSG101 conditional knockout fibroblasts 48 h after infection with a pBabe-Cre virus (*Tsg101^-/-^*) and their isogenic control cells expressing endogenous TSG101 (*Tsg101^fl/fl^*), which were infected with the pBabe control vector. Cells were stained with antibodies against Rab11 (green) and Rabaptin-5 (red); bar represents 10 µm. Arrows indicate examples for a colocalization of both markers (yellow) in wildtype controls.

**Figure 3 cancers-12-00450-f003:**
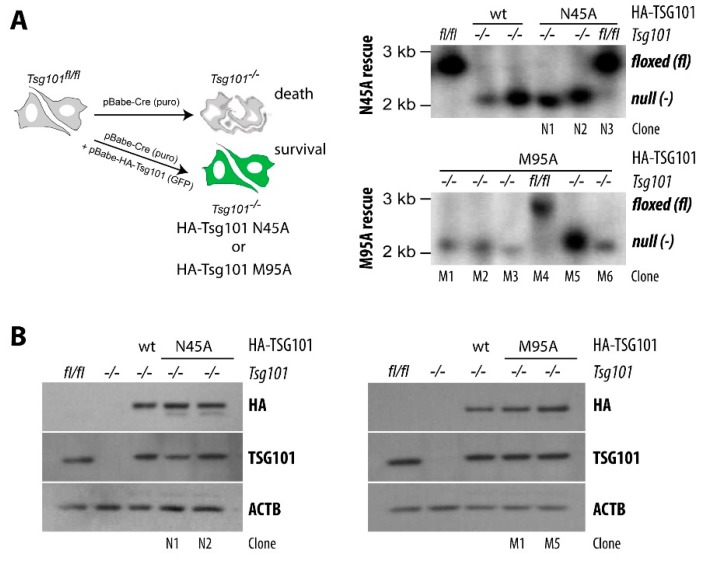
Exogenous wildtype TSG101 and mutants with impaired ubiquitin (N45A) or PTAP (M95A) binding restore growth and survival of mouse fibroblasts that lack both endogenous *Tsg101* alleles. (**A**) Experimental design of genetic rescue experiments in mouse TSG101 conditional knockout cells (left panel) to generate clones that exclusively express HA-tagged, exogenous wildtype or N45A and M95A mutants of TSG101. The right panel shows Southern blot results to verify the complete deletion of both endogenous *Tsg101* alleles in individual rescue clones. (**B**) Western blot analyses to determine the expression of exogenous, HA-tagged TSG101 (wt, wildtype; N45A and M95A mutants) in selected rescue clones in comparison to the parental cells expressing endogenous TSG101 (*Tsg101^fl/fl^*). Detailed methods for the genetic rescue experiment, Southern blot strategy, and immunoblot reagents and methodologies can be found in our earlier publications [15,38,57].

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
