# Peer review of "The Multifaceted Roles of the Tumor Susceptibility Gene 101 (TSG101) in Normal Development and Disease"

_cancers, 2020, doi:10.3390/cancers12020450_

Round 1
Reviewer 1 Report
The review article by Ferraiuolo et al. covers an extensive list of references on the role of Tsg101 gene. As the authors pointed out in the manuscript, there are more than 600 papers on Tsg101. However, a clear consensus on how this versatile factor exerts its actions has not been reached. The manuscript is very well organized and comprehensive, covering various topics and controversies surrounding Tsg101. It is pleasant to read and will help many researchers in the field to grasp the information on Tsg101 in one place.
I have a couple of minor suggestions.
1. Lines 452-572: Move this information to lines 200-204.
2. Section 10 (Summary and Outlook): Much of it contains redundant remarks. I suggest shortening it considerably with a concluding summary.
Author Response
Thank you for taking the time and effort to review our manuscript. We are glad that the reviewer felt that our manuscript is very well organized, comprehensive, and pleasant to read. As the reviewer pointed out, this review might help other researchers in the field to grasp the information on Tsg101 in one place, which was our primary intent.
The reviewer suggested a couple of minor changes. As requested, we considerably shortened the final section and re-organized the Summary and Outlook section into a succinct Summary of one paragraph with concluding remarks. We also considered moving the information provided in lines 452ff to 204ff. Upon close inspection, however, the content did not fit. The earlier section discusses only the phenotypes of knockout models in the context of normal development, whereas the other section focuses solely on oncogenic roles of TSG101 and abnormal phenotypes in models where TSG101 was overexpressed. The main sections were organized by phenotypic consequences of the loss- or the gain-of-function and tumor suppressor versus oncogenic functions and not by organ-specific roles of TSG101.
Thank you again for your valuable suggestions.
Reviewer 2 Report
In this review, “The multifaceted roles of the Tumor Susceptibility 1 Gene 101 (TSG101) in normal development and disease” by Ferraiuolo et al, the authors present a timely discussion of an aspect of the biology of this broadly-studied protein that has been controversial and elusive, i.e., its function as a negative growth regulator and tumor suppressor. The authors raise the legitimate point that the intracellular functions proposed for Tsg101, particularly those in endocytic trafficking, that are often linked to cancer and neurodegeneration, are typically not validated in genetically defined in vivo disease models or in primary human tissue samples. They provide examples where the phenotypes of defined Tsg101 genomic knockout models deviate from observations in cell lines where Tsg101 is only knockdown with siRNAs (CRISPR/Cas-based targeted knockouts or other gene editing methods) have not been widely employed to date). They conclude that it is not yet clear that the acknowledged molecular mechanisms, derived from studies conducted in tissue culture, are responsible for the functions of the protein linked to proliferation and survival. The authors provide a comprehensive review of the question that reinforces the acknowledged gaps. However, the discussion provides little new insight and some parts where unpublished observations from the authors’ own studies are used to challenge the peer-reviewed published literature seem inappropriate to include.
Comments
1. Line 186; line 209; line 262; line 274; line 304; line 346; line 420: There are many instances in the text where the authors expand/extend/ context published results with their own unpublished findings. While any single one of the instances cited might be permitted, collectively they are unacceptable in a literature review because there are so many.
2. Lines 282-307, Figure 3, lines 511-514: These sections of the text describe unpublished findings that should be subjected to peer review prior to being used as a basis for assessing the validity of links between cell survival and Tsg101-participation in RTK ubiquitination/trafficking. This is especially the case as the authors describe several findings in the literature as invalid, inconclusive, tenuous and controversial. As this literature has (presumably) undergone review, it is inappropriate for the authors of this review to challenge literature conclusions with unpublished findings that have not been similarly validated.
3. Line 300-301: “….rescue clones did not exhibit a significant upregulation of Hrs to compensate for the suggested weaker binding affinity”
It does not necessarily follow that Hrs would be upregulated from reduced Hrs-Tsg101 interaction through PSAP motif. Hrs contains at least two additional determinants of Tsg101 interaction and, if Tsg101 is functioning in the context of ESCRT-I, Vps37 also contains interaction sites for Hrs.
Minor Comments
1. Abstract, line 19: “TSG101 a tumor suppressor”. Please edit to “TSG101 as a tumor suppressor”.
2. Line 140: “the main mode of (miss) directing”…... Please edit to “the main mode of (mis)directing”…..
3. The quality of the panels in Figure 3 need to be improved: Text is blurred, blots are fuzzy, some have been heavily cropped, there are no molecular weight markers.
4. Line 371: Please edit “exists” to “exist”.
5. Line 397: “Only slightly more than 200 papers among a total of 676 articles on TSG101 in PubMed can be 397 retrieved in a search for ‘TSG101’ and ‘cancer’ to date, but virtually all publications on TSG101 reference its suggested role as a tumor susceptibility gene in cancer.” Please update this number to 333 of 682. Also, unless the authors have inspected all of the papers, it is suggested that they remove their claim that all publications on Tsg101 reference its suggested role as a tumor susceptibility gene in cancer.
Author Response
The authors thank the reviewer for taking the time to read our manuscript and for providing suggestions for improvement. The reviewer gave an excellent summary of the most important issues that were discussed in our manuscript.
The three main comments were all related to the discussion of unpublished observations that our team made over more than a decade. In the revised version of the manuscript, we have excluded almost the entire description of observations related to the analysis of endosomal markers from confocal microscopy studies as well as experiments related to the internalization and routing of EGF and EGFR to the endosome and lysosome. This seems appropriate since we did not provide any figures. We have also deleted three western blot panels in Fig. 3 (C-E) that showed the expression of HRS and other ESCRT proteins in the knockout-rescue cell lines. We have also deleted the discussion of the omitted data in several other parts of the manuscript, such as how the true knockout of TSG101 affected the levels of Vps28. When the discrepancies between the TSG101 and VPS28 knockouts in the fly are being discussed, we only cite the published work of other teams. We are pleased that this journal accepts the inclusion of a limited number of new findings that are important for the general discussion. This manuscript has undergone peer-review, and any data that is shown as part of figures should not be considered as unpublished observations. Statements about additional experiments that were done by our team do not contradict the current literature and are meant to highlight the potential validation of specific findings made by our colleagues (e.g. the role of TSG101 in recycling). Showing that particular TSG101 mutants are able to rescue the deleterious phenotypes of a complete knockout of TSG101 (Fig. 3) is important for this review to highlight fundamental gaps in our knowledge, to put the previously proposed cellular functions into perspective, and to underline the importance that suggested intracellular roles should be validated using genetically defined models. Any of our knockout mice, derived cell lines, and knockout-rescue models are available to everyone, and we are happy to also share the new tools that we show in this manuscript.
Minor Comments
Abstract, line 19: “TSG101 a tumor suppressor”. Please edit to “TSG101 as a tumor suppressor”. DoneLine 140: “the main mode of (miss) directing”…... Please edit to “the main mode of (mis)directing”….. Done
The quality of the panels in Figure 3 need to be improved The figure has undergone a major revision. As mentioned above, the immunoblot images of three panels were deleted.
Line 371: Please edit “exists” to “exist”.
Done
Line 397: “Only slightly more than 200 papers among a total of 676 articles on TSG101 in PubMed can be 397 retrieved in a search for ‘TSG101’ and ‘cancer’ to date, but virtually all publications on TSG101 reference its suggested role as a tumor susceptibility gene in cancer.” Please update this number to 333 of 682. Also, unless the authors have inspected all of the papers, it is suggested that they remove their claim that all publications on Tsg101 reference its suggested role as a tumor susceptibility gene in cancer.
We deleted the entire paragraph.